# communications
# earth & environment

# Shifts in flood generation processes exacerbate regional flood anomalies in Europe

Larisa Tarasova[1✉], David Lun[2], Ralf Merz[1,3], Günter Blöschl[2], Stefano Basso[1,4], Miriam Bertola[2], Arianna Miniussi[1], Oldrich Rakovec[5,6], Luis Samaniego[5], Stephan Thober[5] & Rohini Kumar[5]

Anomalies in the frequency of river floods, i.e., flood-rich or -poor periods, cause biases in flood risk estimates and thus make climate adaptation measures less efficient. While observations have recently confirmed the presence of flood anomalies in Europe, their exact causes are not clear. Here we analyse streamflow and climate observations during 1960-2010 to show that shifts in flood generation processes contribute more to the occurrence of regional flood anomalies than changes in extreme rainfall. A shift from rain on dry soil to rain on wet soil events by 5% increased the frequency of flood-rich periods in the Atlantic region, and an opposite shift in the Mediterranean region increased the frequency of flood-poor periods, but will likely make singular extreme floods occur more often. Flood anomalies driven by changing flood generation processes in Europe may further intensify in a warming climate and should be considered in flood estimation and management.

[1] Department Catchment Hydrology, Helmholtz Centre for Environmental Research – UFZ, Halle (Saale), Germany. [2] Institute of Hydraulic Engineering and Water Resources Management, Vienna University of Technology, Vienna, Austria. [3] Institute of Geosciences and Geography, Martin-Luther University Halle-Wittenberg, Halle (Saale), Germany. [4] Norwegian Institute for Water Research (NIVA), Oslo, Norway. [5] Department Computational Hydrosystems, Helmholtz Centre for Environmental Research – UFZ, Leipzig, Germany. [6] Faculty of Environmental Sciences, Czech University of Life Sciences Prague, Prague-Suchdol, Czech Republic. ✉email: larisa.tarasova@ufz.de

There is clear evidence for the occurrence of flood-rich and flood-poor periods worldwide from historical and systematic streamflow observations[1–5]. During flood-rich periods, floods of large magnitudes consistently occur more often than usual, and the opposite is true during flood-poor periods[4]. Such flood anomalies may result in biased estimates of the future flood hazard[6,7], thus compromising the reliability of flood risk management measures. Estimating the flood hazard from data during a flood-rich period may result in excessive costs[8], while flood-poor periods may reduce flood preparedness by provoking a false sense of security, leading to unexpected disasters[9–11]. The mechanisms underlying flood anomalies are still poorly understood[12], but believed to be related to the dynamics of the large-scale atmospheric and oceanic circulation[2,13] and corresponding temporal clustering of extreme rainfall[14,15]. Moreover, there are historical evidences that in a warming climate atmospheric drivers of flood anomalies might have changed from atmospheric circulation variability to increasing water vapour[16].

In a changing climate not only the dynamic processes of the climate system but also the flood generation processes on the land surface are changing[17–20]. Observed shifts in the timing of floods within the year and process analyses suggest that, in cold regions, floods caused by rain-on-snow are becoming more frequent at the expense of snowmelt floods, while in other regions, convective events become more frequent at the expense of synoptic events[21–23]. These shifts in flood generation processes affect the magnitude of individual flood events, their spatial extent and synchronicity[24–26].

Given the role of flood generation processes on the land surface and extreme rainfall in controlling the magnitude of individual flood events[25,27], we hypothesise that they also affect the occurrence of flood anomalies, which are even more relevant to flood hazard estimation and capture more complex patterns of flood changes than provided by the trend analysis[12]. However, flood anomalies have so far not been explicitly linked to changes in flood generation processes and extreme rainfall, yet understanding this linkage would contribute to more reliable predictions and flood management.

Here we combine observed flood anomalies in Europe with a flood process typology and show that shifts in the flood generation processes on the land surface have contributed more to the occurrence of regional flood anomalies than changes in extreme rainfall. More so as this study leverages on strictly observational datasets of flood occurrences and their peak magnitudes, it provides greater confidence on underlying analysis and complements ongoing modelling initiatives on flood assessment[28–31].

We base our analysis on the annual maxima of the observed streamflow series of 1353 European catchments for the period from 1960 to 2010 taken from the European Flood Database[21,32]. We identify flood anomalies as unusually frequent (i.e., flood-rich periods) or infrequent (i.e., flood-poor periods) than expected exceedances by annual maxima of three thresholds corresponding to 2-, 5- and 10-year return periods using a method based on scan statistics[3].

The flood events associated with the streamflow peaks are classified into four flood generation processes: (1) floods generated by rainfall on dry soils (Rain.Dry), (2) floods generated by rainfall on wet soils (Rain.Wet), (3) floods generated by simultaneous rainfall and snowmelt (Rain.Snow) and (4) floods generated only by snowmelt (Snowmelt) based on information on the hydrometeorological drivers, soil moisture and snow[33,34]. Linear trends in the frequency of these processes in each catchment are estimated by Sen's slope[35] and the significance of monotonic trends is evaluated by the Mann-Kendall test[36].

We compare the probability distributions of flood magnitudes stratified by flood generation processes in four European regions (Northern, Atlantic, Mediterranean and Central-Alpine, Supplementary Fig. 3) by a pairwise two-sided Kolmogorov-Smirnov test. We also compare the differences in the frequency of flood generation processes during regionally prevailing flood-rich periods with those during flood-poor periods by a $\chi^2$ test.

We assess the significance of a changing frequency of flood generation processes and changing extreme precipitation (i.e., 1-day and 7-day annual precipitation maxima) for explaining flood anomalies in the four regions by comparing a binomial generalised linear model[37] that accounts for these controls with a baseline model that does not via a likelihood ratio test. The contribution to the fit of the models of both controls is quantified by a general dominance measure[38].

We find that the frequencies of flood generation processes in Europe have changed distinctly during 1960–2010 (Fig. 1) and differ significantly between regionally prevailing flood-rich and flood-poor periods. Changing flood generation processes are significant in explaining the occurrence of flood-rich and flood-poor periods and their contribution is higher than that of extreme precipitation in most European regions. This suggests that the ongoing changes in flood generation processes may further exacerbate the occurrence of flood anomalies which should be accounted for in flood estimation and management as climate change advances.

## Results

**Changing frequency of flood generation processes.** Our data show that the frequency of floods generated by rainfall on wet soils (Rain.Wet) mostly increases in the Northern and in Atlantic regions (Fig. 2a) with respectively 45% and 34% of catchments exhibiting significant positive trends (Figs. 1b and Fig. 2b), while the frequency of floods generated by rainfall on dry soils (Rain.Dry) decreases in these regions (Fig. 2a). In contrast, the frequency of Rain.Dry floods increases in the Mediterranean and in the Central-Alpine regions (Fig. 2b) with 31% and 35% of the catchments showing positive significant trends (Figs. 1a and 2b). The frequency of snow-impacted (Rain.Snow and Snowmelt) floods clearly decreases with respect to the frequency of the other processes (Supplementary Fig. 8) with 24% to 48% of significant negative trends across the four European regions (Figs. 1c, d and 2b).

**Regional flood hazard of different processes.** The probability distributions of flood discharge magnitudes are an indicator of the flood hazard[8]. Due to variable probability distributions of event rainfall volumes, rainfall intensities, soil moisture and snowpack (that are key components of the resulting flood generation processes) across different locations (see Supplementary Note 7 and Supplementary Figs. 11 and 12), the corresponding probability distributions of flood discharge magnitudes associated with different processes might vary regionally as well. In the Northern region, snow-induced events are associated with on average higher discharge magnitudes compared to other flood generation processes (Fig. 3a). On the other hand, Rain.Wet events, which have become more frequent in recent decades (Fig. 2), are associated with on average smaller magnitudes (Fig. 3a). In the Central-Alpine region, rainfall-induced floods (Rain.Dry and Rain.Wet) are more likely to exhibit extreme magnitudes than other flood processes as shown by the more pronounced right tails of the distributions (Fig. 3d, h). In the Atlantic and in the Mediterranean regions, Rain.Wet floods on average produce higher magnitudes than the other processes, Rain.Dry processes tend to generate smaller floods (Fig. 3b, c). However, the distribution of the Rain.Dry events in the Mediterranean region has an even more pronounced tail than the

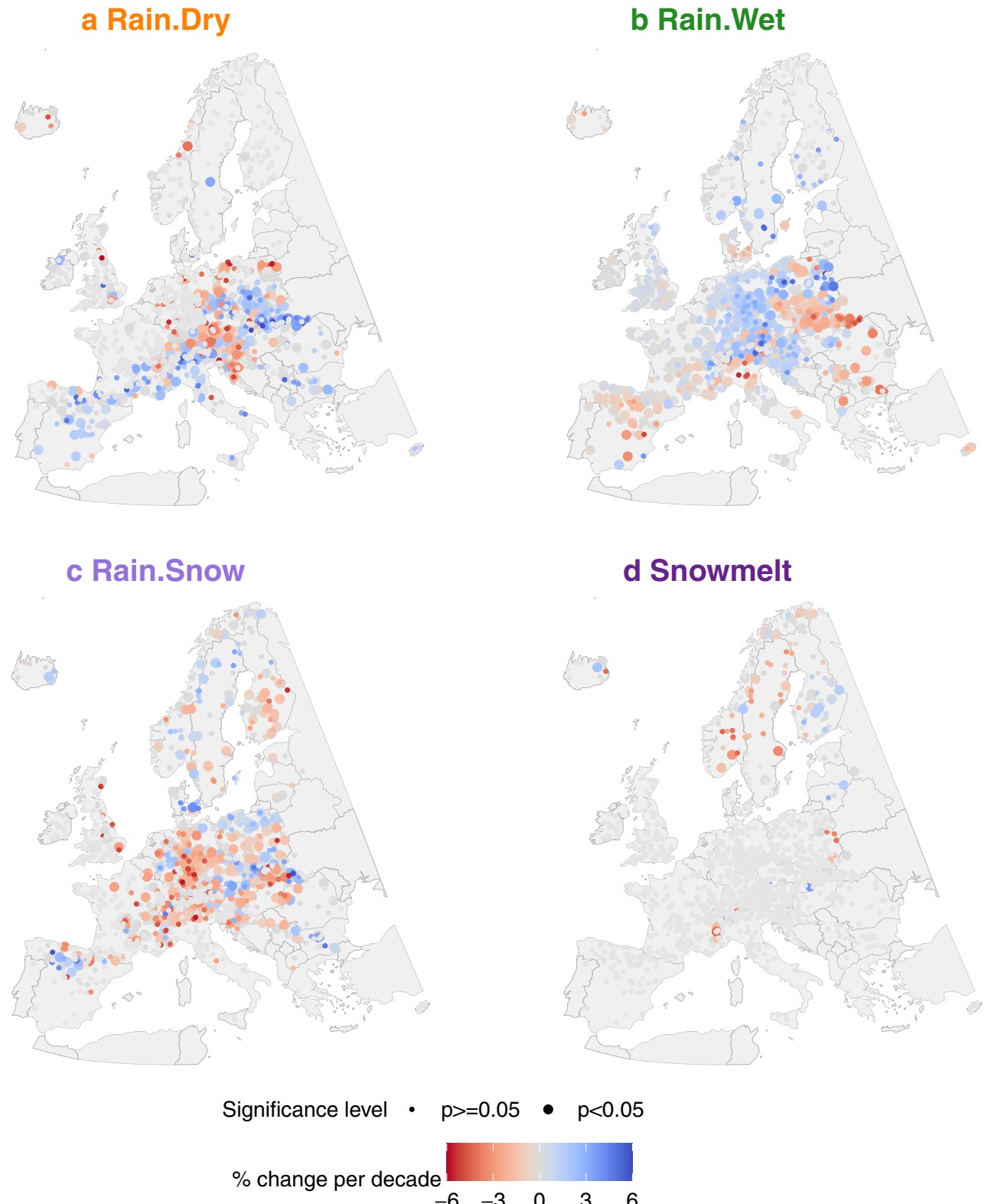

**Fig. 1 Changes in the frequency of flood generation processes in individual catchments in Europe. a–d** Mean change in the frequency [%] per decade over the period 1960-2010 of flood generation processes estimated using Sen's slope (**a** Rain.Dry, **b** Rain.Wet, **c** Rain.Snow and **d** Snowmelt). Catchments with the significant changes (Mann-Kendall test, $\alpha = 0.05$) are indicated as points with larger size. Only catchments with at least five flood events generated by a respective process were considered for trend analysis.

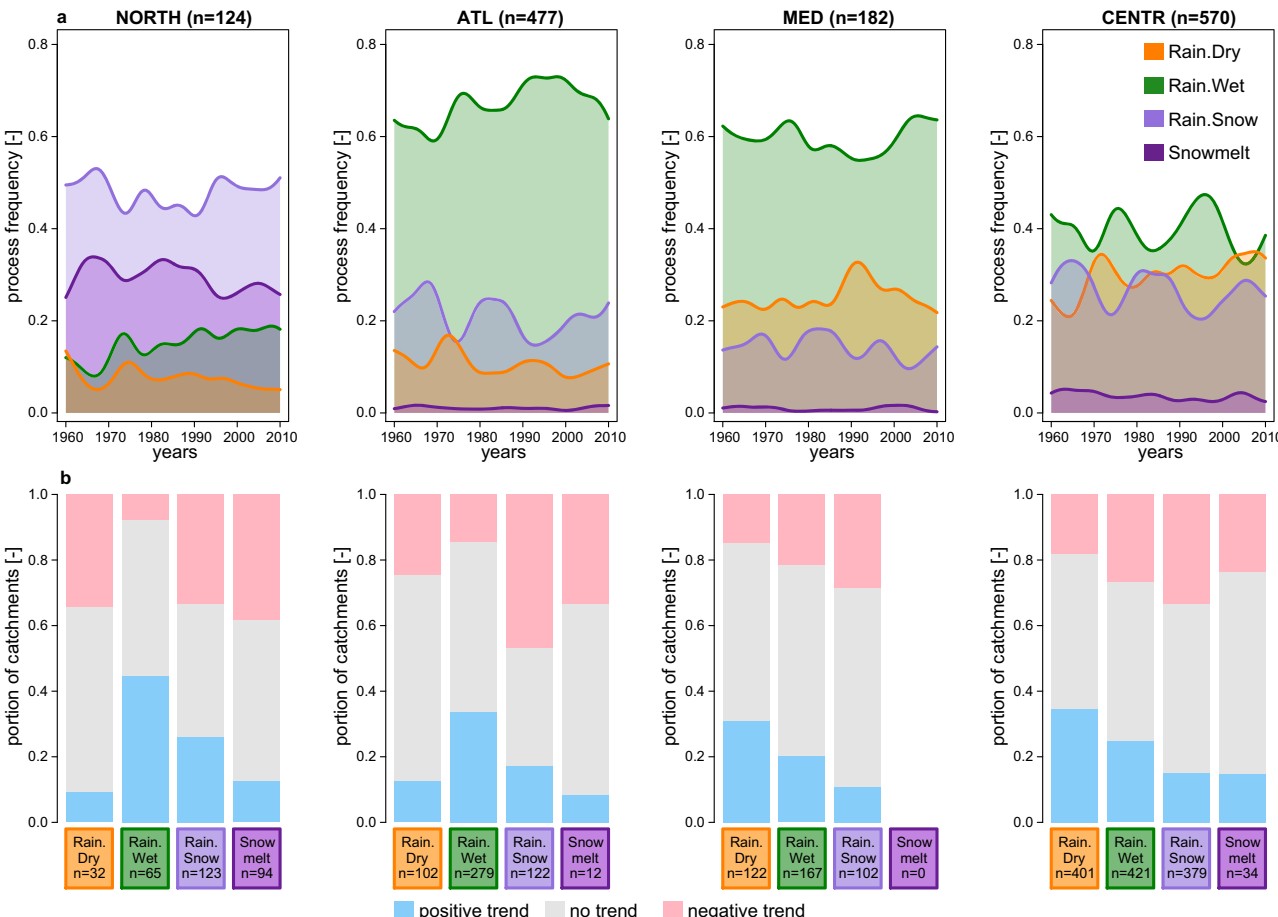

**Fig. 2 Changes in flood generation processes in four European regions. a** Changing annual frequency (smoothed using a kernel with a bandwidth of 7 years) of flood generation processes in the four European regions considered in this study: Northern (NORTH), Atlantic (ATL), Mediterranean (MED) and Central-Alpine (CENTR) (see Supplementary Fig. 3 for the location of the gauges) from 1960 to 2010. $n$ is the number of catchments considered in each region. **b** Portion of catchments in each of the four regions with significant positive and negative trends (Mann–Kendall test, $\alpha = 0.05$) in the frequency of each flood generation process (see Fig. 1 for trends in individual catchments). $n$ is the number of catchments in each region that were analysed for each flood generation process, this number varies because only catchments with at least five flood events generated by a respective process were considered for trend analysis (see "Methods" section).

distribution of Rain.Wet events (Fig. 3g), indicating that they have been responsible for the largest floods on record despite drier antecedent soil conditions.

**The effect of flood generation processes on flood anomalies.** 33% of all study catchments exhibited at least one flood anomaly. The highest portion of catchments exhibiting at least one anomaly was detected in the Mediterranean region (40.1%) and the lowest in the Northern region (27.4%). In the Atlantic and Central-Alpine regions, respectively 33.3% and 31.5% of catchments had at least one flood anomaly. The large floods are clustered in time as evidenced by the presence of flood-rich periods in Fig. 4. The observations show that the frequency of flood-rich and flood-poor periods exhibits a very clear temporal pattern in the four European regions. In the Northern and Mediterranean regions, flood-rich anomalies were more frequent at the beginning of the observation period starting from 1960s, with 9% and 15% of catchments respectively affected by flood-rich anomalies in this period (Fig. 4a). Around the 1980s the situation changed and increasingly more flood-poor anomalies have occurred since then, with on average 15% and 20% of catchments affected during this period in the Northern and Mediterranean regions respectively (Fig. 4a). In the Atlantic

region a rather short, but pronounced (up to 18% of catchments affected), period of flood-poor anomalies occurred in 1970–1980, which was followed by an increasing number of flood-rich anomalies reaching 9% of catchments in 2000 (Fig. 4a). In the Central-Alpine region, two periods with slightly increased frequencies of flood-rich anomalies (up to 7% of catchments affected) were interrupted by prevalent flood-poor anomalies in 1980–1990s with 17% of catchments reporting flood-poor anomalies during this period (Fig. 4a).

In the Atlantic region, a shift in prevailing flood-poor to flood-rich anomalies is aligned with a shift from a period with significantly more frequent Rain.Dry events towards a period with more frequent Rain.Wet events compared to the whole study period by about 5% (Fig. 4b). In the Mediterranean region, a shift in prevailing flood-rich to flood-poor anomalies is aligned with a reverse shift in frequency of Rain.Wet and Rain.Dry events by about 2% (Fig. 4b), which is consistent with lower average magnitudes associated with Rain.Dry compared to Rain.Wet floods in this region (Fig. 3c).

In the Northern region, a shift in prevailing flood-rich to flood-poor anomalies is aligned with a shift from a period with significantly more frequent Snowmelt events towards a period with more frequent Rain.Wet events compared to the whole study period by about 2.5% (Fig. 4b), which is due to the fact that

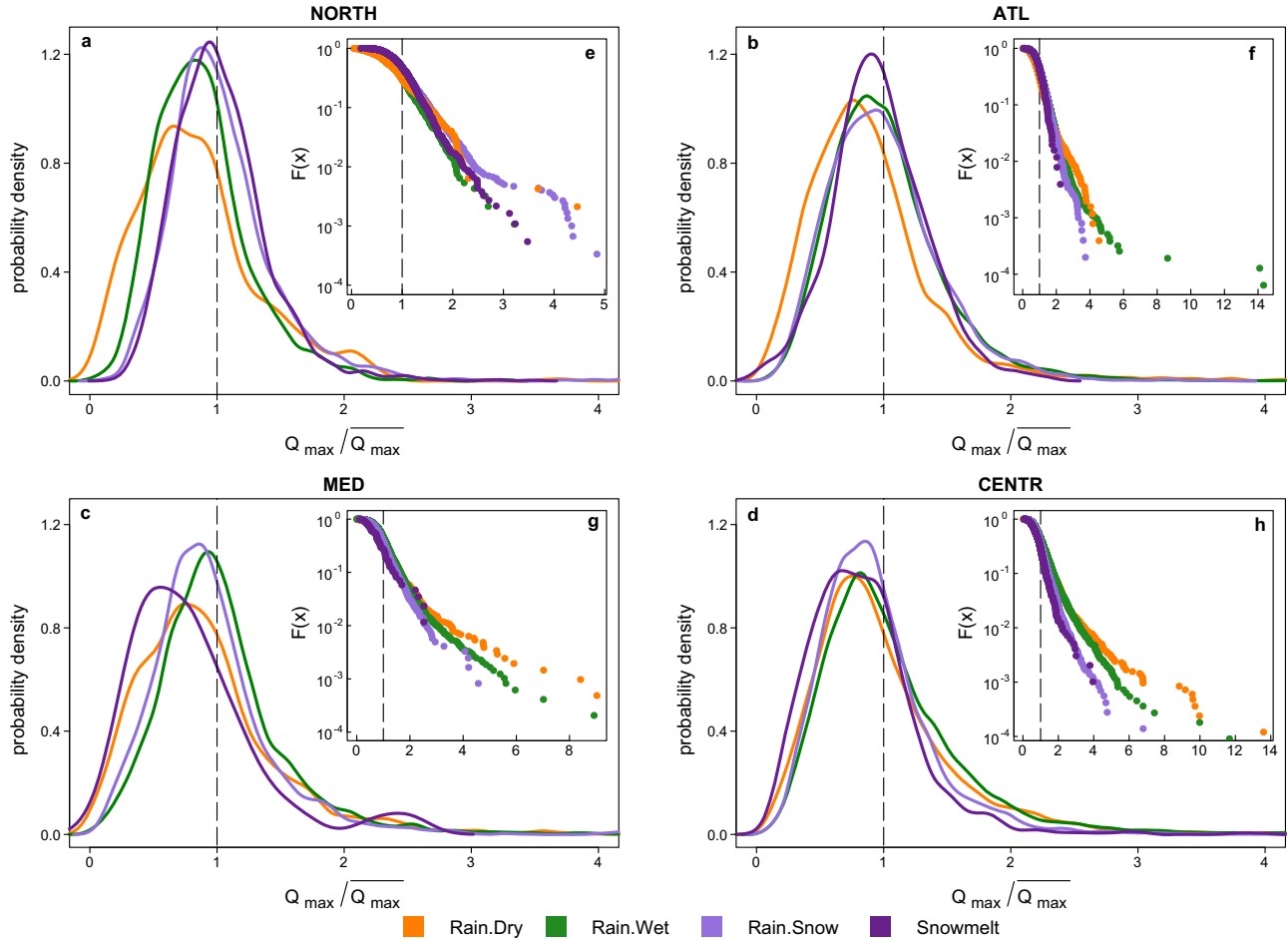

**Fig. 3 Regional probability distributions of flood magnitudes of different flood generation processes. a–d** Kernel density estimates of flood magnitudes of annual maxima Qmax scaled by mean maximum annual flood of all flood events for each study catchment) aggregated for each region and flood generation process. The right tails of probability density functions are only shown up to the value of 4 (four times the mean maximum annual floods) and displayed in full in the corresponding insets. In all panels, the value of 1 on x-axes corresponds to the mean maximum annual flood. The significance of flood magnitude distributions differences for different generation processes is tested with the two-sided Kolmogorov–Smirnov test ($\alpha = 0.01$) with a correction for pairwise comparisons by using a procedure based on the false discovery rate (see Supplementary Table 1). In the NORTH only the differences between the distributions of Snowmelt and Rain.Snow flood magnitudes are not significant. In the ATL only the distribution of Rain.Dry floods is different from all others. In the CENTR and in the MED regions the distributions of magnitudes are significantly different among all four flood generation processes apart from the pair Rain.Dry and Snowmelt floods in the latter region. **e–h** Semi-log plots of the exceedance probability of flood magnitudes (F(x), calculated using Weibull plotting position of floods individually for each generation process) for all study catchments visualising particularly the right tails of regional distributions. The extent of x axes depends on the range of the recorded flood magnitudes in the catchments of corresponding region.

Snowmelt events are associated with larger flood magnitudes than Rain.Wet events in this region (Fig. 3a). In the Central-Alpine region, the period dominated by flood-poor anomalies is characterised by an elevated frequency of Rain.Snow events and fewer rainfall-induced events (Fig. 4b), while during the two periods with a high number of flood-rich anomalies the rainfall-induced events (Rain.Dry and Rain.Wet), which have a more pronounced tail of their flood magnitude distributions (Fig. 3d), are more frequent (Fig. 4b). The differences in flood generation processes during periods dominated by the anomalies of the opposing mode are even more pronounced when only catchments that exhibit at least one anomaly are considered (Supplementary Fig. 9b).

The frequencies of flood generation processes and extreme (7-day maximum) precipitation are significant explanatory variables for the occurrences of flood-poor anomalies (three asterisks in Fig. 5b and Table 1) in all regions, and for the occurrence of flood-rich anomalies in the Northern, Atlantic and Mediterranean regions (Fig. 5a and Table 1). These results indicate that

future changes in the frequency of flood generation processes and the changes in extreme precipitation will significantly affect the occurrence of flood anomalies.

The analysis of general dominance shows that changes in the flood generation processes have higher explanatory power for regional flood-rich anomalies than extreme precipitation in all regions except for the Atlantic, where the contributions are almost equal (bars in Fig. 5a). Similarly, for explaining flood-poor anomalies, flood generation processes have a higher explanatory power in all but the Central-Alpine region, where the contribution of extreme precipitation is higher (Fig. 5b). If one uses 1-day precipitation maxima instead of 7-day maxima the contribution of precipitation is consistently smaller in all cases (Supplementary Fig. 13).

## Discussion and conclusions

This study provides a comprehensive data-based analysis of the effect of changing flood generation processes on the occurrence of

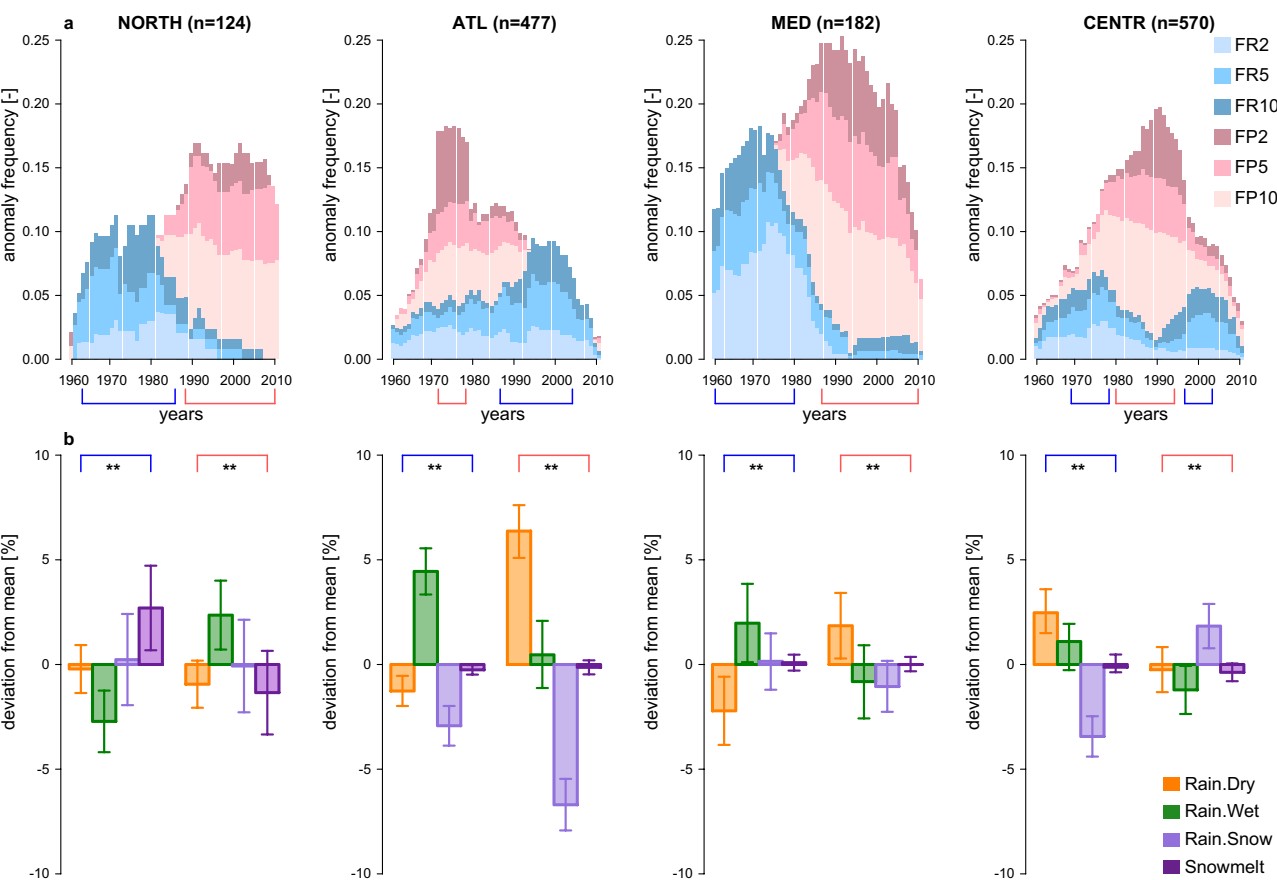

**Fig. 4 Regional flood anomalies and shifts in flood generation processes in four European regions. a** Regionally aggregated flood-rich and flood-poor periods. The height of bars corresponds to the portion of catchments where at least one flood-rich (blue colours) or flood-poor (red colours) anomaly was detected. Flood-rich anomalies are plotted in front of flood-poor anomalies (see Supplementary Fig. 9a for a stacked version). Proportions of anomalies with different return periods (2, 5 and 10 years) are indicated correspondingly FR2, FR5 and FR10 (shades of blue) for flood-rich periods and FP2, FP5 and FP10 (shades of red) for flood-poor periods. **b** Deviations of frequencies of flood generation processes during periods with opposing prevailing anomalies (corresponding to the blue and red time windows indicated on the bottom of the panel a, see also Methods and Supplementary Note 5 for the sensitivity analysis of the definition of these time windows on the test results) compared to the mean composition of the whole study period in all catchments. Error bars show confidence intervals of differences for each flood generation process for a significance level of $\alpha = 0.05$. The significance of these differences is evaluated using a $\chi^2$ test with false discovery rate correction among regions (** indicates significant differences with $\alpha = 0.05$ of false discovery rate correction; exact p-values for each individual $\chi^2$ test are provided in Supplementary Table 2) for each period with opposing prevailing anomalies indicated by the blue and red time windows.

regional flood anomalies at the continental scale. While previous studies have identified the effects of flood generation processes on magnitudes and spatial extents of individual floods in Europe[25,32], we show that the underlying processes also related to the temporal clustering of floods quantified by anomalies of flood-rich and flood-poor periods. These results complement existing analyses that have associated flood anomalies with large scale atmospheric oscillations[2,13]. We also show that the flood generation processes play an even more dominant role in modulating the occurrence of flood anomalies than extreme precipitation in most European regions. The regional consistency of the detected flood anomalies suggests that changing climate and land surface processes not only modulate flood probabilities in single catchments but at the continental scale.

The decreasing frequency of snow-impacted floods (Rain.Snow and Snowmelt) found here across all European regions is consistent with local and regional floods studies[23,39] and evidence on decreasing snow depths and snow water equivalents driven by increasing temperatures mainly in the Northern[40], but also in the Atlantic, Central-Alpine[41] and in the Mediterranean regions[42]. The decrease of Snowmelt floods, which tend to be large, in the Northern region is likely to continue with increasing warming[40],

which will further intensify the current flood-poor period in this region.

The increase of frequency of Rain.Dry events in the Mediterranean and in the Central-Alpine regions is aligned with increasing evaporation trends, which imply a decrease in soil moisture[43]. In the Mediterranean region, where floods are usually associated with high soil moisture[44,45], a further increase of evaporation[46] and hence an increase in the relevance of Rain.Dry events will likely enhance the intensity of the current regional flood-poor period. However, the distributions of the discharge magnitudes of the Rain.Dry events have a more pronounced tail than those of the other processes (Fig. 3), and the distributions do not seem to change over time (Fig. 6; Supplementary Fig. 10, Supplementary Table 3 and Supplementary Note 6). Consequently, if the frequency of Rain.Dry events will continue to increase, a more pronounced tail in the total distribution can be expected, which implies that the probability of singular extreme events will likely increase. This is also in line with the findings that despite mostly negative trends observed in flood magnitudes in the Mediterranean region the magnitudes of large floods (i.e., 100-year return period) decreased less compared to smaller floods[47], and was recently termed as the worst of both world's

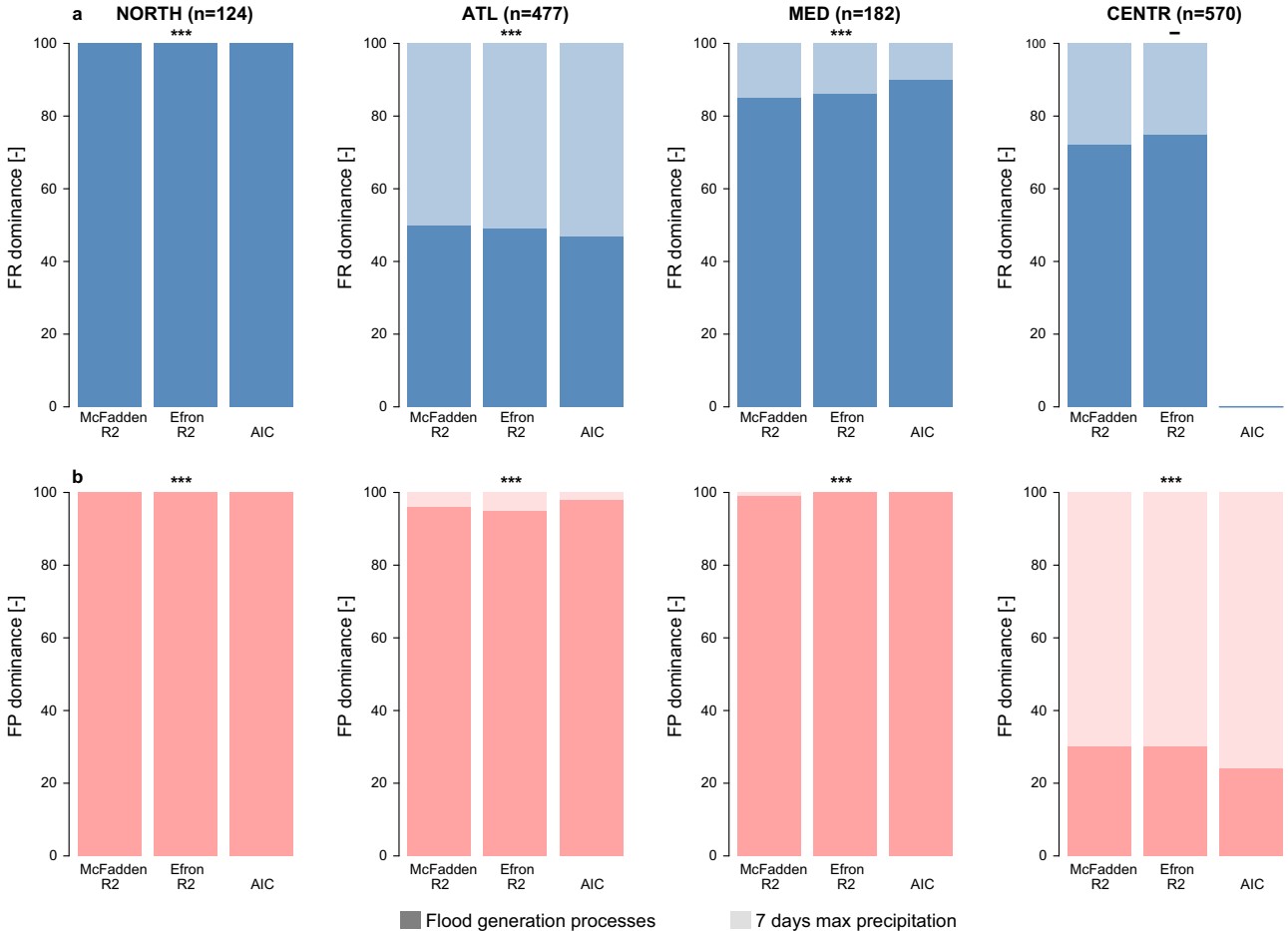

**Fig. 5 General dominance (additional contributions) of temporal variation in flood generation processes and extreme precipitation for predicting the probability of occurrence of regional flood anomalies using binomial generalised linear models.** Importance of flood generation processes and 7 days precipitation maxima for flood anomalies (**a** flood-rich periods (FR); **b** flood-poor periods (FP)) in terms of various performance metrics (McFadden $R^2$, Efron $R^2$ and Akaike Information Criterion (AIC), see Table 1). Higher value of dominance indicates higher contribution to the respective performance measure. The dominance in terms of the AIC is only displayed if the full model (i.e., including the flood generation processes and the extreme precipitation) improves in terms of information content compared to the baseline model that only uses the intercept (i.e., $AIC_{baseline}-AIC_{full}$ is higher than 2, see Table 1). Significance of the covariates ($p < 0.01$ (***), $p < 0.05$ (**), $p < 0.1$ (*), $p > 0.1$ (-)) is provided for each region and anomaly on the top of the corresponding plot. n indicates the number of catchments considered in each region.

scenarios which puts strain on both water availability and flood risk[48].

In the Central-Alpine region, Rain.Dry floods have become more frequent, aligned with more frequent flood-rich periods, and Rain.Snow floods have become less frequent (Fig. 2 and Supplementary Fig. 8a). This shift is mainly related to shallower snow packs rather than to changes in precipitation because there are no consistent trends in summer precipitation observed in this region[49], while seasonal snowpacks have clearly become shallower[41]. Nevertheless, future projected increases of intense summer precipitation[20,50] will likely increase the frequency of Rain.Dry floods in Central-Alpine region, which are often associated with low soil moisture in summer, thus further exacerbate the chance of flood-rich periods and increase flood hazards.

In the Atlantic region the frequency of Rain.Wet floods, which are typically associated with wet winter conditions[32,51], increases consistently with the detected increase of winter precipitation[45,52]. A projected increase of winter precipitation[53] points towards an exacerbation of the current regional flood-rich period in this region.

Our research advances the current understanding of the regional sensitivity of floods to the ongoing climate change and

complements existing flood modelling studies[28–31] providing observational evidences. The analysis of observed streamflow data in 1353 European catchments suggests that, in the South and the North of Europe, prevailing flood-rich periods have shifted to prevailing flood-poor periods in the last 50 years, while the opposite is the case in the Northwest of Europe. During the same period, flood generation processes on the land surface have changed. In the South, events caused by rainfall on dry soils have become more frequent at the expense of rainfall on wet soils, while the opposite is the case in the Northwest. In the North, rain floods have become more frequent at the expense of snowmelt floods. We show that the shift in flood generation processes is a significant control of the occurrence of the flood-rich and flood-poor anomalies. If the flood generation processes continue to change in the future in accordance with current regional projections[20,40,46,50,53], the occurrence of flood anomalies will be exacerbated.

An intensification of regional flood-rich periods in the Northwest will reduce the reliability of existing flood management infrastructure and increase adaption costs. Given that the anomalies occur at a regional scale, there is an elevated risk for floods to co-occur in multiple river basins[25,26], which may

**Table 1 Performance of binomial generalised linear models predicting the probability of occurrence of flood anomalies in each region using portions of flood generation processes and different precipitation indices as explanatory variables.**

| | NORTH ($n = 124$)[e] | | ATL ($n = 477$) | | MED ($n = 182$) | | CENTR ($n = 570$) | |
|---|---|---|---|---|---|---|---|---|
| | FP | FR | FP | FR | FP | FR | FP | FR |
| **7-days precipitation max** | | | | | | | | |
| Likelihood ratio test[a] | $p < 0.01$ | $p < 0.01$ | $p < 0.01$ | $p < 0.01$ | $p < 0.01$ | $p < 0.01$ | $p < 0.01$ | $p = 0.18$ |
| Pseudo $R^2$ (McFadden)[b] | 0.08 | 0.05 | 0.06 | 0.15 | 0.04 | 0.03 | 0.06 | 0.02 |
| Pseudo $R^2$ (Lave/Efron)[c] | 0.14 | 0.11 | 0.09 | 0.32 | 0.09 | 0.04 | 0.09 | 0.04 |
| $AIC_{baseline}-AIC_{full}$[d] | 29.54 | 11.65 | 60.52 | 68.43 | 19.39 | 14.75 | 57.05 | −1.15 |
| **1-day precipitation max** | | | | | | | | |
| Likelihood ratio test | $p < 0.01$ | $p < 0.01$ | $p < 0.01$ | $p < 0.01$ | $p < 0.01$ | $p < 0.01$ | $p < 0.01$ | $p = 0.40$ |
| Pseudo $R^2$ (McFadden) | 0.08 | 0.05 | 0.06 | 0.12 | 0.05 | 0.03 | 0.01 | 0.02 |
| Pseudo $R^2$ (Lave/Efron) | 0.14 | 0.11 | 0.09 | 0.27 | 0.09 | 0.05 | 0.02 | 0.05 |
| $AIC_{baseline}-AIC_{full}$ | 29.54 | 11.65 | 68.19 | 54.97 | 22.10 | 17.23 | 12.65 | 0.03 |

[a]Significance of the covariates (i.e., frequencies of flood generation processes and precipitation maxima for a given year in each region).
[b]An analogy of $R^2$ of linear regression models for generalised linear models based on log-likelihood[72].
[c]A correlation-based analogy of $R^2$ of linear regression models for generalised linear models[72].
[d]AIC describes the trade-off between the goodness of fit and the simplicity of the model. Higher AIC values indicate higher information losses by the model[74]. Differences in Akaike Information Criterion ($AIC_{baseline}-AIC_{full}$) that are >2 indicate considerable improvement of the model[76] accounting for flood generation processes and precipitation maxima compared to the baseline model (i.e., only uses intercept and assumes that the probability of occurrence of a flood anomaly in each given year equals the mean observed probability of the corresponding flood anomaly in the whole study period).
[e]In the Northern region model coefficients of both precipitation covariates are equal to zero, resulting in identical performance of the models including 7-days and 1-day precipitation maxima.

challenge emergency response and reduce socio-economic resilience[54,55]. An intensification of flood-poor periods in the South and North may indirectly increase flood risk by provoking a false sense of security and deteriorating flood preparedness[9,10]. This could be especially the case in the South, where the processes favouring flood-poor periods increase the probability of singular extreme floods[11].

The exacerbation of regional flood anomalies calls for regionally coordinated flood risk estimation and management plans. Estimation must be grounded on processes-based methods, given the effect of changing flood generation processes on the occurrence of flood anomalies found in this study.

## Methods

**Flood event database and causative classification of flood events.** In this study, we use the European flood database[21,32], which contains information on the date and the maximum peak discharge of observed annual maximum floods in 2370 European catchments for the period from 1960 to 2010. Corresponding beginning and end points of each reported flood event are extracted from daily streamflow time series simulated by the well-established mHM model[56,57] using an automated event identification method[58] (see Supplementary Note 2). The beginning and end points of events are used to attribute flood-inducing precipitation and snowmelt and to identify antecedent soil moisture prior to each flood event (i.e., one day before the start of the flood event). Precipitation and temperature information for classification of flood events is obtained from a downscaled 5 km E-OBS product[59,60]. Snowmelt and soil moisture that are also used for classification are simulated using the mHM model[56,57]. Indicators derived from these data are used to classify all events using the process-based framework for event characterisation and classification[34] (see Supplementary Note 4 and Supplementary Fig. 7 for the list of indicators and their corresponding classification thresholds) into the following four groups of flood generation processes: (1) flood events generated by rainfall on dry soils (Rain.Dry), (2) flood events generated by rainfall on wet soils (Rain.Wet), (3) flood events generated by simultaneous rainfall and snowmelt (Rain.Snow) and (4) flood events generated purely by snowmelt (Snowmelt).

**mHM simulations.** The mHM model[56,57] is driven by a downscaled 5 km E-OBS product[59] over the European domain for the period 1960–2010 at daily resolution. The model uses a seamless Multiscale Parameter Regionalization (MPR) parameterisation scheme[60] and was previously cross-validated across 357 European catchments[61]. Modelled fluxes and states are simulated on a 5 km grid, and streamflow is routed by the multiscale routing algorithm[62].

We link catchments from the European Flood Database to the mHM grid using the outlet coordinates and area of catchments provided in the database resulted in 1444 catchments for which the databases were deemed consistent (Supplementary Fig. 1, Supplementary Note 1). Among the 1444 mHM-linked catchments, 1353 catchments fit the requirements of the anomaly detection procedure[3] (i.e., at least 40 years of maximum annual flood observations between 1960 and 2010, with the record starting in 1968 or earlier, and ending in 2002 or later). The median size of the selected catchments is 349 km[2] (Supplementary Fig. 2).

Model performance is evaluated in terms of model ability to simulate observed peak discharges and timing of maximum annual floods. Spearman rank correlation $r$ [-] used to assess model performance in terms of flood magnitudes gives mean r values of 0.53–0.60 for all study catchments for three different comparison cases (Supplementary Fig. 5a) which is considered suitable for the purposes of this study, given that instantaneous peak flows are compared with daily averages and the simulated streamflow is not used directly for the analysis of flood magnitudes. Performance in terms of timing is evaluated by the absolute difference in [days] between the observed annual flood and simulated annual flood/corresponding runoff event (see Supplementary Note 3 and Supplementary Figs. 4–6 for details on methodology and model performance).

**Regional relevance and temporal changes of flood generation processes.** All catchments are aggregated into four regions that correspond to the major geographical regions in Europe and are associated with the distinct seasonality of floods[21]: Northern (124 catchments), Atlantic (477 catchments), Mediterranean (182 catchments) and Central-Alpine (570 catchments) (Supplementary Fig. 3). The Northern region includes snow-dominated catchments of the Scandinavian Peninsula and the Baltics that are associated with summer and spring floods. The Atlantic region includes Western European catchments that instead are associated with winter floods[21]. The Mediterranean region also includes catchments that are most exclusively have winter floods and located in the Southern Europe. Finally, the Central-Alpine region mostly includes catchments with summer and spring floods that cover the Alps and Carpathian Mountains[21]. We identify the regional relevance (i.e., frequency) of each flood generation process for each year (i.e., number of catchments where a given flood generation process caused a maximum annual flood in a given year) for the whole study period 1960–2010.

To visualise possible temporal changes in the relevance of flood generation processes, we aggregate the frequency of the four processes for each decade from 1960 to 2010 and for each region (Fig. 2a). To quantify temporal changes in the frequency of each of the four flood generation processes and to determine the significance of these changes in the period 1960–2010 we perform a non-parametric trend analysis using Sen's slope to estimate the magnitude of the linear trend[35] and the exact Mann–Kendall test[36] to test the significance level of the monotonic trends ($\alpha = 0.05$) for each catchment (Fig. 1), which is a standard practise in hydrological studies[63]. To ensure the robustness of the results we only perform the trend analysis if at least 5 events were generated by the flood generation process of interest in the analysed catchment (in total 3971 events (i.e., 5%) were discarded from trend analysis). The minimal number of events for each type was selected as a trade-off between the desired robustness of trend analysis and data availability (classification of 51 year of annual maxima into four flood generation processes will result in 12-13 events for each process assuming uniform distribution of frequency of process occurrence)[25]. These events were not discarded from any further quantitative analysis. The results of catchment-wise trend analyses (Fig. 1) are aggregated to the four regions (Fig. 2b) by summarising the

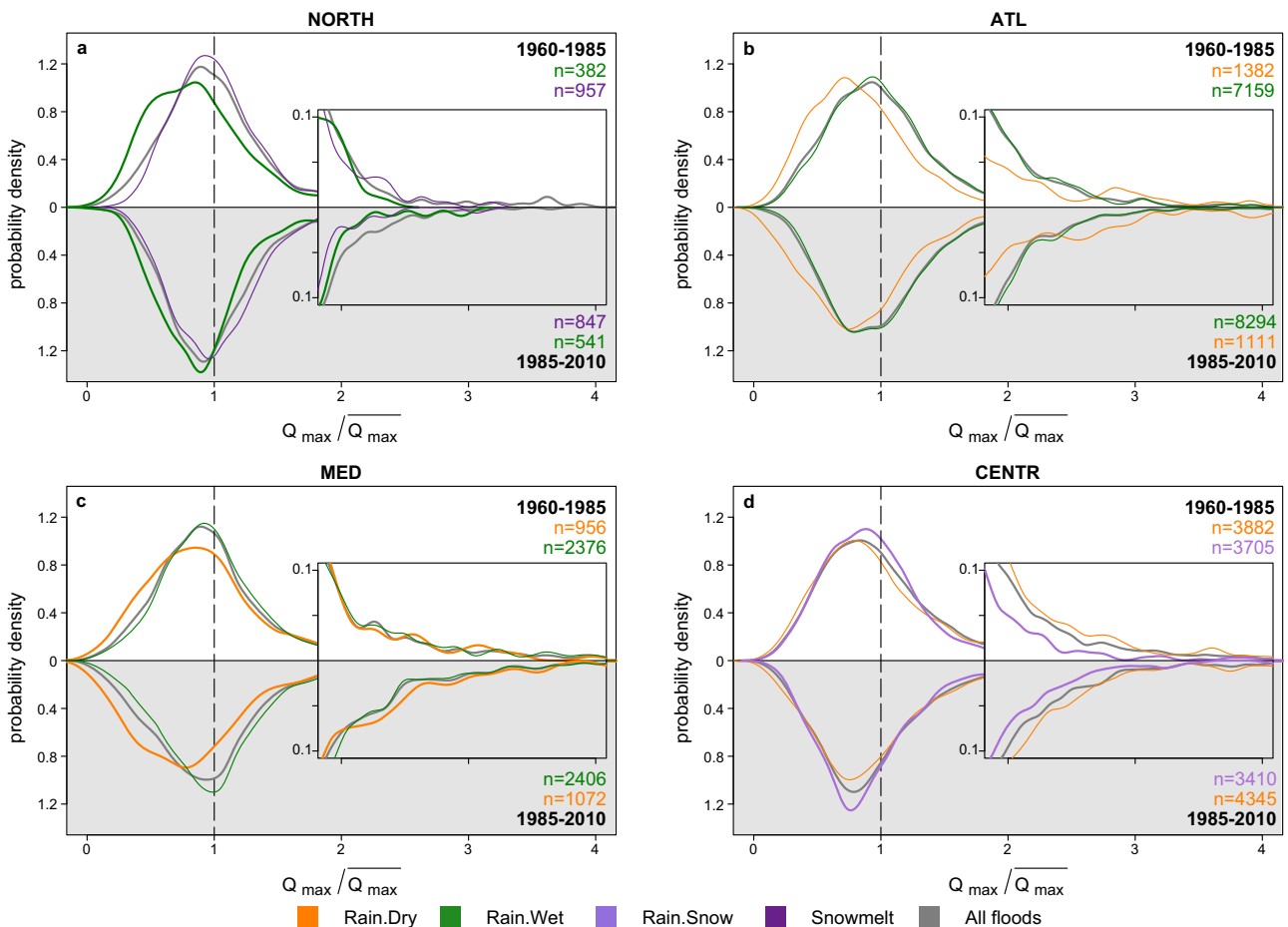

**Fig. 6 Comparison of regional flood probability distributions of flood generation processes aggregated for two different periods. a–d** Regional kernel density estimates of flood magnitudes (Qmax scaled by mean maximum annual flood for each study catchment) aggregated for each region and flood generation process. Insets provide a zoom into the right tails of corresponding distributions. The right tails of probability density functions are only shown up to the value of 4 (four times the mean maximum annual floods). The significance of differences of flood magnitude distributions of flood generation processes aggregated for two different periods (1960-1985 (white background) and 1985-2010 (grey background)) is tested with the pairwise two-sided Kolmogorov-Smirnov test ($\alpha = 0.01$) applying a false discovery rate correction. Significant cases are displayed as thick lines. Differences between different flood generation processes within the same period are significant for all displayed cases ($\alpha = 0.01$) according to the pairwise two-sided Kolmogorov-Smirnov test with false discovery rate correction (see Supplementary Table 3). $n$ is the number of flood events in each sample. Only flood generation processes that showed considerable changes in their regional frequency (Fig. 2) and have considerable deviations in their frequency during opposing modes of flood anomalies (Fig. 4b) are displayed. The remaining processes are displayed in Supplementary Fig. 10.

number of catchments within each region with significant positive and significant negative trends.

### Regional probability distributions of flood magnitudes for each flood generation process.
Regional probability distributions of observed flood magnitudes of four flood generation processes are estimated by scaling the series of annual flood peaks by their corresponding mean values in each catchment and pooling all catchments in a region, stratified by flood generation processes, and applying a kernel density estimator (Fig. 3a–d). The significance of differences between the regional distributions of flood generation processes is evaluated using a two-sided Kolmogorov-Smirnov test with a correction for pairwise comparisons using the procedure based on the false discovery rate[64] (Supplementary Table 1). Additionally, we derived regional exceedance probability distributions and display them as semi-logarithmic plots to highlight the behaviour of the right tails of distributions that correspond to the largest flood events in the region.

### Detection of flood anomalies and their regional aggregation.
Flood-rich periods (FR) of a flood series are defined here as coherent periods in time when the frequency of the observed annual floods peaks exceeding a certain threshold is higher than expected[3]. Flood-poor periods (FP) are defined as coherent periods in time when the frequency of annual flood peaks not exceeding a certain threshold is higher than expected. This study analyses flood anomalies previously detected in Europe in the study of Lun et al.[3] that considers three different thresholds: estimates of flood magnitudes that correspond to 2-, 5- or 10-year return periods[3]. The

procedure to identify flood anomalies for the time-discrete series of annual maximum floods is based on scan statistics[65]. The reference condition for the detection procedure is defined using the assumption that annual maximum flood peaks can be modelled with independent identically distributed (iid) random variables, which implies that threshold exceedances follow a time-homogeneous Bernoulli process. The number of threshold exceedances is counted for all possible coherent windows of observations with a specified length (see Lun et al.[3] for details) and windows with unusually many, respectively few, exceedances are identified as flood-rich, respectively flood-poor, periods, if they are statistically significant. The significance of temporal clusters is evaluated with a scan statistic (which corresponds to a maximum of exceedances over all possible coherent windows of observations). Large values of the scan statistic (corresponding to unlikely clusters of threshold exceedances under to the iid-assumption) correspond to a statistically significant generalised likelihood ratio test[66] of a time-homogeneous Bernoulli-process vs a time-varying Bernoulli-process. A significance level of $\alpha = 0.05$ was used for all flood-rich and flood-poor anomalies reported here.

Detected flood anomalies are aggregated regionally as the number of catchments within each of the four regions (i.e., the Northern, Atlantic, Mediterranean and Central-Alpine) that report at least one flood-rich (flood-poor) period in a given year within the study period regardless of the investigated threshold (i.e., flood magnitudes corresponding to 2-, 5- or 10-year return periods) (Fig. 4a, the height of the bars). Since, for a given year, a catchment can simultaneously have several anomalies with regard to different thresholds (e.g., with regard to 2-, 5- or 10-year return periods), we also compute the total number of anomalies per year and per region and corresponding proportion of anomalies

with regard to the three investigated thresholds (Fig. 4a, colour-coding for FR2, FR5, FR10 and FP2, FP5, FP10). It is worth to note that in this study we do not estimate field significance of flood anomalies, as its estimation is not straightforward for test statistics with a discrete distribution[67] and existing approaches are not suitable for multivariate test statistics.

**The difference in the composition of flood generation processes during regional flood anomalies**. In order to detect possible differences in the composition of flood generation processes, we compare the frequency of the four flood generation processes during periods dominated by the opposing modes of flood anomalies (i.e., flood-rich and flood-poor periods) with the mean frequency of flood generation processes for the whole study period (Fig. 4b). Due to an expectedly lower amount of flood-rich compared to flood-poor anomalies[3] periods with regionally-prevailing flood-rich anomalies are defined as uninterrupted time windows when at least 5% of catchments are affected by flood-rich anomaly and there are twice as many catchments affected by flood-rich anomalies than by flood-poor anomalies in a given region (Fig. 4a, blue time windows). Accordingly, periods with regionally-prevailing flood-poor anomalies are defined as uninterrupted time windows when the frequency of flood-poor anomalies (i.e., portion of affected catchments) exceeds the frequency of flood-rich anomalies at least 2.5 times and flood-rich anomalies are detected in less than 5% of catchments in a given region (Fig. 4a, red time windows). The effect of this definition on the results of this study is examined in a sensitivity analysis by varying the limits (i.e., varying the starting and ending years up to 3 years) of identified regionally-prevailing flood-rich and flood-poor anomalies (see Supplementary Note 5). The sensitivity analysis results show that the significance of differences in the frequency of flood generation processes during opposing modes of anomalies is insensitive to the choice of the particular starting and ending point of the time window (Supplementary Table 2) and the above-mentioned definition of time windows is further used in the analysis (Fig. 4).

Confidence intervals of differences for each flood generation process are calculated via a normal approximation for differences in proportions for a significance level of $\alpha = 0.05$. The significances of the detected differences (i.e., the difference in the composition of flood generation processes during time periods dominated by the opposing modes of anomalies compared to the whole study period) are evaluated with a $\chi^2$ test for each anomaly and each region separately using false discovery rate correction to account for multiple hypothesis testing (Fig. 4b).

**Extreme precipitation**. To compare the effect of changing relevance of flood generation processes to the effect of changing extreme precipitation, we extract 1-day and 7-days annual maximum catchment-averaged precipitation series from a downscaled 5 km E-OBS product[59] and further average them over the four regions for each year (see Supplementary Fig. 14).

**Binomial generalised linear models**. To evaluate the effect of changing flood generation processes and event precipitation on the probability of occurrence of regional flood anomalies we fit binomial regressions formulated as generalised linear models[37] with a logit link function without accounting for possible spatial correlations[68]. This procedure models the probability of occurrence of a given flood anomaly (i.e., flood-rich or flood-poor) in a given year in a given region as a binomial distribution where the yearly portions of catchments reporting a corresponding flood anomaly is the response variable. Accordingly, portions of catchments with floods of each of four flood generation processes and extreme precipitation are used as explanatory variables excluding the intercept to account for mutually exclusive and collectively exhaustive flood generation processes. 1-day and 7-day precipitation maxima are used individually one at a time to construct two models for each region and anomaly to avoid any potential issues with multicollinearity. It is worth to note that the detection of flood anomalies and classification of each flood according to its generation processes are two independent procedures. Model parameters for precipitation covariates are restricted to only allow physically plausible relation with flood anomalies: for flood-rich anomalies coefficients of precipitation covariates are required to be higher or equal to zero and for flood-poor anomalies, the coefficients are required to be less or equal to zero. This restriction is applied, because in the Northern region fitting of the unrestricted binomial generalised linear model for flood-rich anomalies resulted in negative coefficients for extreme precipitation (i.e., for both 7-days and 1-day precipitation maxima), indicating that an increase of precipitation leads to a decreased probability of occurrence of flood-rich anomalies and increased probability of occurrence of flood-poor anomalies. A clear increase of precipitation in the Northern region (Supplementary Fig. 14) and a corresponding decrease in the frequency of flood-rich anomalies since 1980 (Fig. 4a) highlights the lack of a causal link between these two phenomena in this region. Parameter restriction is a common practise, especially for the case of Bayesian models, to avoid spurious correlation and increase physical realism of statistical models[45]. All model parameters are estimated using the R package 'glmnet'.

The significance of the explanatory variables (i.e., flood generation processes and extreme precipitation) is evaluated using a likelihood ratio test. The explanatory power of the fitted models is evaluated by McFadden R[2] [69], which is a log-likelihood based Pseudo-R[2] and Lave/Efron R[2] [70,71], which is a correlation-based Pseudo-R[2] for generalised linear models (i.e., an alternative for a common R[2] used for linear models)[72]. The values of Pseudo-R[2] are typically lower than the R[2] of linear regressions. Usually Pseudo R[2] cannot take on the value 1 and the range of values 0.2-0.4 can be considered an excellent fit in practice[73]. We evaluate the additional value of the models containing the information on the time-dependent regional frequency of flood generation processes compared to a baseline model that only uses the intercept, or in other words assumes that the probability of occurrence of a flood anomaly in each given year equals the mean observed probability of the corresponding flood anomaly in the whole study period. We use the Akaike Information Criterion (AIC)[74] that in addition to evaluating the goodness of fit of a model also penalises more complex models proportionally to the number of model parameters.

**Dominance analysis of model covariates**. To assess the contribution of the covariates (i.e., flood generation processes and extreme precipitation) to the explanatory power of the fitted binomial generalised linear model a dominance analysis is performed[38]. We compute the measure of general dominance that reflects an average improvement in the goodness-of-fit measure (i.e., Pseudo-R[2] or AIC) when a covariate of interest is included in the model for all possible model subsets. For the interpretation of the results, we focus on a normalised general dominance measure[75] that is computed by dividing the general dominance measure of each covariate by the corresponding goodness-of-fit measure of the full model (i.e., Pseudo-R[2] or decrease of AIC). The normalised general dominance measures of all covariates sum up to unity. Normalised general dominance of the covariates for model improvement in terms of information content (i.e., AIC) is only computed if the full model that includes both flood generation processes and extreme precipitation improves (i.e., its AIC decreases) compared to the baseline model (i.e., $AIC_{baseline} - AIC_{full} > 2$)[76].

## Data availability

The European Flood database that provides date and peak discharges of annual maximum floods is available from[21] and[32] (https://github.com/tuwhydro/europe_floods). Input data (downscaled precipitation and temperature for the E-OBS) and model simulations (snow water equivalent and soil moisture) used for flood classification can be obtained from EDgE Project (http://edge.climate.copernicus.eu/). Processed data (classified flood series and series of regional anomalies) are deposited in https://doi.org/10.5281/zenodo.6403851.

## Code availability

The code for the performed analytical analysis and producing the main figures of this study can be found in https://doi.org/10.5281/zenodo.6403860.

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

## Acknowledgements

The financial support of the German Research Foundation ("Deutsche Forschungsgemeinschaft", DFG) in terms of the research group FOR 2416 "Space-Time Dynamics of Extreme Floods (SPATE)" and the research project 421396820"Propensity of rivers to extreme floods: climate-landscape controls and early detection (PREDICTED)", the Austrian Science Fund ("Fonds zur Förderung der wissenschaftlichen Forschung", FWF) in terms of subproject I 3174 and the Helmholtz Centre for Environmental Research (UFZ) is gratefully acknowledged. R.K., O.R., S.T. and L.S. also acknowledge the funding from Copernicus Climate Change Service (edge.-climate.copernicus.eu). We thank Lina Stein and two anonymous reviewers for their helpful comments that have improved the original manuscript.

## Author contributions

L.T., D.L., R.M. and G.B. designed the study. L.T. wrote the first draft of the paper with the contributions from D.L., R.M. and G.B.; L.T. conducted analysis of flood generation processes with the contributions from S.B. and A.M.; D.L. identified flood anomalies. L.T. and D.L. conducted the analysis of the effects of flood generation processes on flood anomalies. M.B. assisted with preparation of the regional flood data. R.K., O.R., S.T. and L.S. provided mHM simulations and the downscaled hydrometeorological data. R.K. and O.R. linked catchments to the mHM grid. All authors contributed to framing and revising the manuscript.

## Funding

## Competing interests

The authors declare no competing interests.
