## [Peer Review File · Communications Earth & Environment]

18th Jan 23

Dear Dr Tarasova,

Your manuscript titled "Shifts in flood generation processes exacerbate regional flood anomalies in Europe" has now been seen by our reviewers, whose comments appear below. In light of their advice I am delighted to say that we are happy, in principle, to publish a suitably revised version in Communications Earth & Environment under the open access CC BY license (Creative Commons Attribution v4.0 International License).

We therefore invite you to revise your paper one last time to address the remaining concerns of our reviewers. In particular, we would like to encourage you to extend the data analysed beyond the year 2010, if possible (reviewer 2's comment labelled "L286").

At the same time we ask that you edit your manuscript to comply with our format requirements and to maximise the accessibility and therefore the impact of your work.

EDITORIAL REQUESTS:

*****Please take care to match our formatting and policy requirements. We will check revised manuscript and return manuscripts that do not comply. Such requests will lead to delays. *****

SUBMISSION INFORMATION:

In order to accept your paper, we require the files listed at the end of the Editorial Requests Table; the list of required files is also available at <https://www.nature.com/documents/commsj-file-checklist.pdf> .

OPEN ACCESS:

Communications Earth & Environment is a fully open access journal. Articles are made freely accessible on publication under a [CC BY license](http://creativecommons.org/licenses/by/4.0) (Creative Commons Attribution 4.0 International License). This license allows maximum dissemination and re-use of open access materials and is preferred by many research funding bodies.

For further information about article processing charges, open access funding, and advice

and support from Nature Research, please visit https://www.nature.com/commsenv/article-processing-charges

At acceptance, you will be provided with instructions for completing this CC BY license on behalf of all authors. This grants us the necessary permissions to publish your paper. Additionally, you will be asked to declare that all required third party permissions have been obtained, and to provide billing information in order to pay the article-processing charge (APC).

[link redacted]

Best regards,

Yama Dixit
Editorial Board Member
Communications Earth & Environment

Heike Langenberg, PhD
Chief Editor
Communications Earth & Environment

On Twitter: @CommsEarth

REVIEWERS' COMMENTS:

Reviewer #1 (Remarks to the Author):

This study analyzed the influence of flood generation mechanisms on the anomalies of flood occurrence in Europe during 1960-2010. The floods were attributed to their generation mechanisms using a hydrologic model and the magnitude of floods caused by each mechanism were analyzed in four regions of Europe. The study then analyzes the regional changes in the frequency of floods caused by the four mechanisms during 1960-2010 and its impact on the occurrence of flood-rich and flood-poor periods, defined as periods with high and low frequency of floods exceeding a certain threshold in a region. The dominant role of flood generation mechanisms on the occurrence of flood rich or poor periods was confirmed using binomial generalized linear models with extreme rainfall and frequency of flood

generation mechanisms as predictors.

This study addresses a highly relevant issue by trying to understand how changes in flood generation mechanisms contribute to the occurrence of flood rich and poor periods in different regions. This understanding is crucial for more accurate quantification of flood risk and changes in flood risk due to climate change. The manuscript is very well written with concise discussions and good quality figures. However, some clarifications and additional discussions are required regarding the choice of regions, land use characteristics and the role of precipitation in flood magnitudes. The manuscript will be well-suited for publication in *Communications Earth & Environment* after some minor changes. The following are my detailed comments:

1. In the study, the catchments are grouped into four regions: Northern, Atlantic, Mediterranean and Central-Alpine and the subsequent changes in flood generation mechanisms are analyzed at the regional scale. However, the basis for this regional classification is not clear; are these regions with similar climate or catchment features? The choice of the regions will have a significant impact on the results regarding the influence of flood generation mechanisms on the flood magnitudes. For instance, if the catchments belonging to a certain region have similar climates but have different catchment properties such as elevation, topography and land-use, then the flood generation mechanisms within that region cannot be assumed to be homogeneous. The authors should clarify the methodology to select regions in which catchments have similar climate and flood generation mechanisms.
2. The changes in flood generation mechanisms detected in this study are based on the assumption that there are no significant changes in catchment characteristics during the study period. It is not clear how the authors address the change in flood generation mechanisms caused due to human activities such as changes in land-use land cover. Have the authors verified if there are no significant human interventions in the selected catchments?
3. The effect of flood generation mechanism on the flood magnitude is also influenced by the magnitude of rainfall that occurs during the storm. Therefore, whether a rain-dry event will produce a flood with higher or lower magnitude compared to a rain-wet event will depend on how much rainfall occurs within the storm. However, in the manuscript the effects of storm magnitude and flood generation mechanism on the flood magnitude have not been distinguished. This can be done for instance by plotting the probability distributions of rainfall magnitudes in storms associated with floods of different mechanisms similar to Figure 2. Similarly, for analyzing the effect of flood generation mechanism on the flood rich and flood poor periods, the authors can plot the changes in the magnitudes of storms associated with different flood generation mechanisms similar to Figure 3b.

Reviewer #2 (Remarks to the Author):

Title: Shifts in flood generation processes exacerbate regional flood anomalies in Europe

Summary

The authors identify flood-rich and flood-poor periods in Europe. By linking them to flood generating processes the authors are able to explain changes in flood occurrence and offer estimates in regard to future flood expectations.

The authors demonstrate the importance of time-sensitive analysis of flood anomalies. Simply relying on trend detection is not enough, when flood generating processes fluctuate over time. This analysis then takes the our understanding of flood generation changes over time to the next step, by adding flood generating processes to the explanation. Overall, I think that this article offers a novel and detailed look at current past flood behaviour in Europe. The results are presented in high quality (and visually pleasing) figures and support the conclusions drawn. While the authors seem to make data and code available, this review did not take them into account, since access was denied. Besides this, I only have some minor comments and open questions for the authors to address. I would recommend publication in Communications Earth&Environment.

Lina Stein

Comments:

L29: The term “large floods” is ambiguous in the sense that it can either refer to high magnitude or large extend of floods. Consider a different wording here.

L53: I do not agree that this study “leverages on strictly observational flood datasets”, since a big part of the analysis rests on modelled outputs. It might mislead the reader to assume all data is observation based.

L44-49: Either here or later I would add a sentence that mentions the difference and the benefits of a flood anomaly vs a flood trend analysis. There have been quite a few recent flood trend studies, so this distinction is relevant and sets apart this study from other similar ones.

L59: Either here or in the paragraph from L130 onwards, can you extend on how the anomalies are measured regarding the different return periods? One option would be to move the sentence in L338-340 to the introduction.

L68/L307: Does the choice how the catchments are split into different regions have an effect on the results? And, maybe I missed it, but how was this split decided? Especially the differentiation between North and Atlantic catchments is difficult.

L130: Please add how many catchments have detected anomalies in total and how many do not have one. It is mentioned in the methods, but I think it would be good to add here as well.

L200: How do your results compare to results by Brönnimann et al, 2022, who studied atmospheric circulation patterns influence on flood-rich/poor periods in Europe?

L227/261: Wasko et al, 2021 describe the increase of extreme floods combined with a decrease of small floods as the worst of both worlds, since smaller floods usually replenish water storages within the catchment. If you think this is relevant, it could be a nice addition regarding the implications of your findings.

L286: Why did you end the analysis in 2010? There should be another decade of information worth adding, unless there are data constraints?

L319: How many events were discarded because of this criterion? (If this only affects very few events, disregard the next two questions). Where they only excluded for the flood process analysis or also for the flood anomaly calculation? If not included, would they have

significantly changed the classification of flood-poor/flood-rich periods?

L379: I like that you did a sensitivity analysis to test the relevance of the FR/FP time windows. However, I am not sure if ± 3 years is a sufficiently large enough variation to test the sensitivity, if the anomalies can last more than 20 years. Have you thought about varying the definitions themselves to see the impact (e.g. twice as many catchments FR than FP period criterion for example)? Visually the found anomaly periods make sense, though I wonder if the FR period in the Atlantic region does not start a bit early.

Fig2: Is Qmax and mean Qmax process specific? Or is mean Qmax calculated for all events?

Extended Data Figure 3: Please add the legend here as well. Figure 1 is at a completely different place of the manuscript and not easily consulted.

Brönnimann, S., Stucki, P., Franke, J., Valler, V., Brugnara, Y., Hand, R., ... & Schaepli, B. (2022). Influence of warming and atmospheric circulation changes on multidecadal European flood variability. *Climate of the Past*, 18(4), 919-933.

Wasko, C., Nathan, R., Stein, L., & O'Shea, D. (2021). Evidence of shorter more extreme rainfalls and increased flood variability under climate change. *Journal of Hydrology*, 603, 126994.

Reviewer #3 (Remarks to the Author):

(1) Please explain the Scan statistics used for flood-rich and flood-period identifications.

(2) How is the independence of individual flood events assessed? Please explain in the methodology section.

(3) What is the reason behind the exceedingly steep baseflow increase in flood events identifications method?

(4) Why is a 0.7 threshold selected for classifying the flood event caused by snowmelt?

(5) Why is a 0.3 threshold considered for classifying a flood event caused by rain on snow?

(6) Why is 1980 selected as the threshold for the split-sample analysis?

(7) Please explain why increasing the time window by 3 years does not significantly affect the sensitivity analysis results.

(8) A serially correlated dataset can influence the Mann-Kendall (MK) test. Please explain whether serial-correlation corrections are applied in the MK test in the methodology.

(9) Please explain why the flood generation mechanism differs in NORTH, ATL, MED, and CENTR regions. For example, why higher magnitude flood events are produced by rain on wet soil in the MED region?

(10) How do shifts in flood generation mechanisms work differently in the selected regions?

(11) Line 662: the code of the present study is not accessible.

Responses reviewers' comments

The original comments are provided in italic.

Reviewer #1 (Remarks to the Author):

This study analyzed the influence of flood generation mechanisms on the anomalies of flood occurrence in Europe during 1960-2010. The floods were attributed to their generation mechanisms using a hydrologic model and the magnitude of floods caused by each mechanism were analyzed in four regions of Europe. The study then analyzes the regional changes in the frequency of floods caused by the four mechanisms during 1960-2010 and its impact on the occurrence of flood-rich and flood-poor periods, defined as periods with high and low frequency of floods exceeding a certain threshold in a region. The dominant role of flood generation mechanisms on the occurrence of flood rich or poor periods was confirmed using binomial generalized linear models with extreme rainfall and frequency of flood generation mechanisms as predictors.

This study addresses a highly relevant issue by trying to understand how changes in flood generation mechanisms contribute to the occurrence of flood rich and poor periods in different regions. This understanding is crucial for more accurate quantification of flood risk and changes in flood risk due to climate change. The manuscript is very well written with concise discussions and good quality figures. However, some clarifications and additional discussions are required regarding the choice of regions, land use characteristics, and the role of precipitation in flood magnitudes. The manuscript will be well-suited for publication in Communications Earth & Environment after some minor changes. The following are my detailed comments:

We thank the reviewer for a comprehensive summary and for a positive evaluation of our work. Below we respond to the specific comments of the reviewer.

1. In the study, the catchments are grouped into four regions: Northern, Atlantic, Mediterranean and Central-Alpine and the subsequent changes in flood generation mechanisms are analyzed at the regional scale. However, the basis for this regional classification is not clear; are these regions with similar climate or catchment features? The choice of the regions will have a significant impact on the results regarding the influence of flood generation mechanisms on the flood magnitudes. For instance, if the catchments belonging to a certain region have similar climates but have different catchment properties such as elevation, topography and land-use, then the flood generation mechanisms within that region cannot be assumed to be homogeneous. The authors should clarify the methodology to select regions in which catchments have similar climate and flood generation mechanisms.

We apologize for not explaining the assignment of catchments more clearly in the original manuscript. The catchments were assigned to regions similar to the study of Lun et al (2020) to distinguish the main geographical regions in Europe. More specifically, we used the findings of the previous studies on the detection of flood anomalies of Lun et al (2020) and on flood seasonality of Blöschl et al (2017) to derive regions that combine catchments with similar flood seasonality (as a proxy of similarity in flood generation processes) and at the same time show similar patterns of changes in flood anomalies. The main difference between the Atlantic region and the Northern region is the dominance of snowmelt processes in the generation of floods in the latter (Blöschl et al., 2019) and the distinct flood seasonality in these two regions (Blöschl et al., 2017). Major changes in assignments of catchments (e.g., combining catchments with very different flood generation processes) to regions will inevitably affect the outcome, however due to a large number of

catchments assigned to each region the effect from re-assigning single catchments on the results of this study are expected to be limited. We have added a more detailed explanation in the revised manuscript (L. 276-281).

2. The changes in flood generation mechanisms detected in this study are based on the assumption that there are no significant changes in catchment characteristics during the study period. It is not clear how the authors address the change in flood generation mechanisms caused due to human activities such as changes in land-use land cover. Have the authors verified if there are no significant human interventions in the selected catchments?

The dataset of Blöschl et al (2019), that was used in this study, was screened for data errors, and catchments that were known, or were identified, to have experienced strong human modifications (such as reservoirs) that could affect changes in flood discharges were excluded. Indeed, it is not possible to exclude the effect of human interactions in individual catchments in the dataset. However, the analysis of trends in flood generation processes shows that these trends are highly consistent within the same region (Extended Data Figure 2). Regional consistency of these trends gives us confidence that any possible human activities in individual catchments do not affect the results of our regional analysis of the effects of changing flood generation processes on the occurrence of flood anomalies. Moreover, as noted by Blöschl (2022) human interventions tend to have smaller scale impacts on changing floods than climate, which is in line with the consistent continental scale pattern of change found in the Extended Data Figure 2.

Blöschl, G.: Three hypotheses on changing river flood hazards, *Hydrol. Earth Syst. Sci.*, 26, 5015–5033, <https://doi.org/10.5194/hess-26-5015-2022>, 2022.

3. The effect of flood generation mechanism on the flood magnitude is also influenced by the magnitude of rainfall that occurs during the storm. Therefore, whether a rain-dry event will produce a flood with higher or lower magnitude compared to a rain-wet event will depend on how much rainfall occurs within the storm. However, in the manuscript the effects of storm magnitude and flood generation mechanism on the flood magnitude have not been distinguished. This can be done for instance by plotting the probability distributions of rainfall magnitudes in storms associated with floods of different mechanisms similar to Figure 2. Similarly, for analyzing the effect of flood generation mechanism on the flood rich and flood poor periods, the authors can plot the changes in the magnitudes of storms associated with different flood generation mechanisms similar to Figure 3b.

We thank the reviewer for this helpful suggestion. It is indeed interesting to see the distribution of precipitation volumes and intensities for different flood generation processes. We have now added Supplementary Note 7, Figures S8 and S9 that show probability distributions of precipitation volume (rainfall and snowmelt) and maximum precipitation intensity that correspond to different flood generation processes. It can be seen that different flood generation processes are associated with distinct distributions of precipitation properties (e.g., as expected Rain.Dry events that usually occur in summer are associated with the highest intensities and energy-limited Snowmelt events have the smallest intensities). Please notice that Figure 3b only shows differences in flood generation processes during anomaly periods and does not consider flood magnitudes. Therefore, an analogous Figure for event rainfall is not possible. However, the effect of precipitation alone is considered in the binomial model where we compare the effect of flood generation processes and precipitation for the occurrence of flood anomalies using dominance analysis.

Reviewer #2 (Remarks to the Author):

Title: Shifts in flood generation processes exacerbate regional flood anomalies in Europe

Summary

The authors identify flood-rich and flood-poor periods in Europe. By linking them to flood generating processes the authors are able to explain changes in flood occurrence and offer estimates in regard to future flood expectations.

The authors demonstrate the importance of time-sensitive analysis of flood anomalies. Simply relying on trend detection is not enough, when flood generating processes fluctuate over time. This analysis then takes our understanding of flood generation changes over time to the next step, by adding flood generating processes to the explanation. Overall, I think that this article offers a novel and detailed look at current past flood behavior in Europe. The results are presented in high quality (and visually pleasing) figures and support the conclusions drawn. While the authors seem to make data and code available, this review did not take them into account, since access was denied. Besides this, I only have some minor comments and open questions for the authors to address. I would recommend publication in Communications Earth & Environment.

Lina Stein

We thank Lina Stein for positive feedback on our manuscript and for the helpful comments that are addressed below.

We apologize that the code and the data were not accessible during the review process. The permissions were set to make them accessible after the manuscript acceptance. The link is provided in the Data and code availability section.

Comments:

L29: The term "large floods" is ambiguous in the sense that it can either refer to high magnitude or large extend of floods. Consider a different wording here.

Indeed, this term might be confusing. We have clarified it in the revised manuscript (L. 30-31).

L53: I do not agree that this study "leverages on strictly observational flood datasets", since a big part of the analysis rests on modelled outputs. It might mislead the reader to assume all data is observation based.

We apologize for the confusion. Here we meant that we used the strictly observational dataset of flood occurrences and magnitudes (i.e., their corresponding dates and peak discharges). We have clarified this in the revised manuscript (L. 56).

L44-49: Either here or later I would add a sentence that mentions the difference and the benefits of a flood anomaly vs a flood trend analysis. There have been quite a few recent flood trend studies, so this distinction is relevant and sets apart this study from other similar ones.

Thank you for your suggestions, we have now added a corresponding clarification to the revised manuscript (L. 49-50).

L59: Either here or in the paragraph from L130 onwards, can you extend on how the anomalies are measured regarding the different return periods? One option would be to move the sentence in L338-340 to the introduction.

Thank you for your suggestions, we have added a corresponding clarification to the revised manuscript (L. 61-63).

L68/L307: Does the choice how the catchments are split into different regions have an effect on the results? And, maybe I missed it, but how was this split decided? Especially the differentiation between North and Atlantic catchments is difficult.

We apologize for not explaining the assignment of catchments more clearly in the original manuscript. The catchments were assigned to regions similar to the study of Lun et al (2020) to distinguish the main geographical regions in Europe. More specifically, we used the findings of the previous studies on the detection of flood anomalies of Lun et al (2020) and on flood seasonality of Blöschl et al (2017) to derive regions that combine catchments with similar flood seasonality (as a proxy of similarity in flood generation processes) and at the same time show similar patterns of changes in flood anomalies.

The main difference between the Atlantic region and the Northern region is the dominance of snowmelt processes in generation of floods in the latter (Blöschl et al., 2019) and the distinct flood seasonality in these two regions (Blöschl et al., 2017). Major changes in assignments of catchments (e.g., combining catchments with very different flood generation processes) to regions will inevitably affect the outcome, however due to the large number of catchments assigned to each region the effect from re-assigning single catchments on the results of this study are expected to be limited. We have added a more detailed explanation in the revised manuscript (L. 276-281).

L130: Please add how many catchments have detected anomalies in total and how many do not have one. It is mentioned in the methods, but I think it would be good to add here as well.

We have moved these results from the method section as suggested. See L. 117-120.

L200: How do your results compare to results by Brönnimann et al, 2022, who studied atmospheric circulation patterns influence on flood-rich/poor periods in Europe?

We thank the reviewer for bringing our attention to this recent work on multidecadal flood variability. We have included the findings from this reference in the revised manuscript (L.38-40). Notably, the identified anomaly periods in the study Brönnimann et al 2022 for the overlapping periods (after 1960) and for the overlapping regions (Northern and Atlantic) correspond very well to the ones that were identified in our study.

L227/261: Wasko et al, 2021 describe the increase of extreme floods combined with a decrease of small floods as the worst of both worlds, since smaller floods usually replenish water storages within the catchment. If you think this is relevant, it could be a nice addition regarding the implications of your findings.

Thank you for this useful suggestion, we have added this to the L. 195-196 in the revised manuscript.

L286: Why did you end the analysis in 2010? There should be another decade of information worth adding, unless there are data constraints?

Unfortunately, currently there is no possibility to extend the analysis performed in this study beyond 2010. This study is based on the largest observation European flood dataset currently available that took several years to collect. Correspondingly, extending this dataset for an additional decade is a major task that will take well beyond the revision time provided for this manuscript.

L319: How many events were discarded because of this criterion? (If this only affects very few events, disregard the next two questions). Where they only excluded for the flood process analysis or also for the flood anomaly calculation? If not included, would they have significantly changed the classification of flood-poor/flood-rich periods?

In total, we disregard 3971 events (i.e., 5% of all events) from the trend analysis of changes in flood generation processes. Please note that this analysis has an illustrative character to display the general picture of changes in flood generation processes in each individual catchment in Europe. These events are not discarded from any further quantitative analysis (i.e., computation of flood anomalies and binomial model). We have clarified this point in the revised manuscript (L. 293 and L. 297-298).

L379: I like that you did a sensitivity analysis to test the relevance of the FR/FP time windows. However, I am not sure if ± 3 years is a sufficiently large enough variation to test the sensitivity, if the anomalies can last more than 20 years. Have you thought about varying the definitions themselves to see the impact (e.g. twice as many catchments FR than FP period criterion for example)? Visually the found anomaly periods make sense, though I wonder if the FR period in the Atlantic region does not start a bit early.

Indeed, the anomalies can persist for two decades in the Northern and in the Mediterranean regions, but they also can be rather short as for example the flood-poor period in the Atlantic and flood-rich periods in the Central-Alpine regions. In the latter cases selecting longer periods for sensitivity analysis will mean considering a different period altogether that might correspond to a very different proportion of the anomalies in the corresponding region. Since the goal of the sensitivity analysis was to see whether or not the selection of exact starting points affects the outcome of this study and not to evaluate the effect of selecting completely different periods (that expectedly will result in a different outcome of the analysis), we prefer to keep the sensitivity analysis limited to 3 years. We have highlighted the essence of the sensitivity analysis in the revised manuscript in L.357-358, L.361 and to the Supplementary Note 5.

Fig2: Is Qmax and mean Qmax process specific? Or is mean Qmax calculated for all events?

Mean Qmax is calculated using all events for each individual catchment. Qmax corresponds to each individual flood event and is therefore not aggregated. We have clarified this in the revised caption of Figure 2.

Extended Data Figure 3: Please add the legend here as well. Figure 1 is at a completely different place of the manuscript and not easily consulted.

Added.

Reviewer #3 (Remarks to the Author):

(1) Please explain the Scan statistics used for flood-rich and flood-period identifications.

We have added a more detailed description of the scan statistics in the Method Section in L. 322-331. For a more detailed description of the methodology, we refer the reader to Lun et al. (2020), the study which produced the results on flood anomalies used in this manuscript.

(2) How is the independence of individual flood events assessed? Please explain in the methodology section.

We use the data set of Blöschl et al. (2019), who did assess the independence of individual flood events via testing for significant lag-1 autocorrelations in the data. Their findings suggest that no significant serial correlation between the flood peaks can be found, after accounting for the underlying trends in flood magnitudes.

(3) What is the reason behind the exceedingly steep baseflow increase in flood events identifications method?

Baseflow was separated as the first step of the event identification procedure that is based on event runoff coefficients and hence requires separation of the total flow into quickflow and baseflow. We have clarified this in the Supplementary Note 2.

(4) Why is a 0.7 threshold selected for classifying the flood event caused by snowmelt?

This threshold indicates that the most of total precipitation volume consists of snowmelt (i.e., 70%). This threshold aligns well with the range of the thresholds used in previous studies (see Tarasova et al., 2019).

(5) Why is a 0.3 threshold considered for classifying a flood event caused by rain on snow?

This threshold indicates that a considerable portion of total precipitation volume originates from snowmelt (i.e., 30%) and the event cannot be considered as a pure rainfall event. The selection of the threshold is based on extensive analysis in previous studies. Please also refer to our response above.

(6) Why is 1980 selected as the threshold for the split-sample analysis?

We sincerely apologize for the typo that has occurred in the manuscript. We have of course performed split sample analysis for periods 1960-1985 and 1985-2010, because 1985 corresponds to the mid-point of the study period. We have corrected the corresponding labels in now in the Supplementary Note 6, Table S2, Figure S7, and the Extended Data Figure 4.

(7) Please explain why increasing the time window by 3 years does not significantly affect the sensitivity analysis results.

Lack of the significance in the effect on the variable definition of regional anomalies indicates that the differences in flood generation processes during the identified periods of anomalies (regardless of their exact start and end) were significantly different than during the whole study period. We have added this explanation to the Supplementary Note 5.

(8) A serially correlated dataset can influence the Mann-Kendall (MK) test. Please explain whether serial-correlation corrections are applied in the MK test in the methodology.

We fully agree with the reviewer that serial correlation can influence the outcome of a Mann-Kendall test. However, in line with our response to a comment (2) of the referee, for the annual series of flood types in our study we do not expect serial correlation and therefore we do not apply a serial-correlation correction.

(9) Please explain why the flood generation mechanism differs in NORTH, ATL, MED, and CENTR regions. For example, why higher magnitude flood events are produced by rain on wet soil in the MED region?

Theoretically, flood generation processes are associated with a certain probability distribution of flood peaks that should be similar across different locations. However, since each flood generation process can be decomposed in the corresponding probabilities of rainfall intensities, rainfall volumes, soil moisture and snow water equivalents, the resulting distribution of flood magnitudes is inevitably related to the probability distributions of these individual components at different locations. For example, the probability of snow accumulation in the Northern region is much higher than the corresponding probability in the Mediterranean region. That is exactly why it is important to consider regional differences in the probability distributions of flood magnitudes of different flood generation processes (Figure 2). We have clarified this in the revised manuscript (L.100-105 and Supplementary Note 7). We also added Figures S8 and S9 that show contrasting distributions of precipitation properties for different flood generation processes.

(10) How do shifts in flood generation mechanisms work differently in the selected regions?

The differences in the effects of a shift in flood generation processes on flood anomalies across different regions emerge directly from different corresponding probability distributions of flood magnitudes (Figure 2). Please refer also to the comment 9 above.

(11) Line 662: the code of the present study is not accessible.

We apologize that the code and the data were not accessible during the review process. The permissions were set to make them accessible after the manuscript acceptance. The link is provided in the Data and code availability section.